# *FtMYB163* Gene Encodes SG7 R2R3-MYB Transcription Factor from Tartary Buckwheat (*Fagopyrum tataricum* Gaertn.) to Promote Flavonol Accumulation in Transgenic *Arabidopsis thaliana*

**DOI:** 10.3390/plants13192704

**Published:** 2024-09-27

**Authors:** Hanmei Du, Jin Ke, Xiaoqian Sun, Lu Tan, Qiuzhu Yu, Changhe Wei, Peter R. Ryan, An’hu Wang, Hongyou Li

**Affiliations:** 1Panxi Featured Crops Research and Utilization Key Laboratory of Sichuan Province, Xichang University, Xichang 615000, China; tanlu19910222@163.com (L.T.); yuqiuzhu20080724@126.com (Q.Y.); hantanheying@126.com (C.W.); 13795660264@163.com (A.W.); 2Research Center of Buckwheat Industry Technology, College of Life Sciences, Guizhou Normal University, Guiyang 550025, China; gemkll@163.com (J.K.); sunxiaoqianxqs@163.com (X.S.); 3Division of Plant Sciences, Research School of Biology, The Australian National University, Canberra, ACT 2601, Australia; peter.ryan@anu.edu.au

**Keywords:** Tartary buckwheat, flavonoid biosynthesis, flavonol, MYB transcription factor, FtMYB163

## Abstract

Tartary buckwheat (*Fagopyrum tataricum* Gaertn.) is a coarse grain crop rich in flavonoids that are beneficial to human health because they function as anti-inflammatories and provide protection against cardiovascular disease and diabetes. Flavonoid biosynthesis is a complex process, and relatively little is known about the regulatory pathways involved in Tartary buckwheat. Here, we cloned and characterized the *FtMYB163* gene from Tartary buckwheat, which encodes a member of the R2R3-MYB transcription factor family. Amino acid sequence and phylogenetic analysis indicate that FtMYB163 is a member of subgroup 7 (SG7) and closely related to FeMYBF1, which regulates flavonol synthesis in common buckwheat (*F. esculentum*). We demonstrated that FtMYB163 localizes to the nucleus and has transcriptional activity. Expression levels of *FtMYB163* in the roots, stems, leaves, flowers, and seeds of *F. tataricum* were positively correlated with the total flavonoid contents of these tissues. Overexpression of *FtMYB163* in transgenic *Arabidopsis* enhanced the expression of several genes involved in early flavonoid biosynthesis (*AtCHS*, *AtCHI*, *AtF3H*, and *AtFLS*) and significantly increased the accumulation of several flavonoids, including naringenin chalcone, naringenin-7-O-glucoside, eriodictyol, and eight flavonol compounds. Our findings demonstrate that FtMYB163 positively regulates flavonol biosynthesis by changing the expression of several key genes in flavonoid biosynthetic pathways.

## 1. Introduction

Tartary buckwheat (*Fagopyrum tataricum* Gaertn), as an annual herb of the Polygonaceae family, is largely distributed in the Sichuan, Yunnan-Guizhou Plateau, and Tibetan regions of China as well as some other high-altitude, arid mountainous areas [1,2]. Tartary buckwheat is rich in flavonoids, including rutin, quercetin, and other flavonols, and is a potentially important crop for medicine and human health [3]. Flavonoids are one of the largest secondary metabolites in plants, and their content is approximately 1.0~3.0% in Tartary buckwheat, most of which resides in the flowers, leaves, and seeds [4,5]. So far, a total of 93 flavonoids have been identified in Tartary buckwheat seeds, and 32 of them are predicted to play important roles in the resistance to major diseases (cancers/tumors, hypertension, and cardiovascular disease) [6]. In addition, 61 flavonoid components have been absolutely quantified in Tartary buckwheat seeds [7].

Flavonoids encompass a wide range of different phenolic compounds, including anthocyanins, flavonols, flavones, and proanthocyanidins. Different flavonoids have different biological functions, and many of these compounds have antioxidant properties that protect plants from biotic and abiotic stresses [8,9]. Flavonoids also play an important role in plant growth and physiology, including cell wall synthesis [10], root growth and phototropism [11,12], and pollen development [13]. Understanding the regulation of flavonoid biosynthesis and metabolism more fully may enable the composition and concentration of flavonoids to be manipulated to enhance the value of Tartary buckwheat as a commercial crop.

Biosynthetic pathways of flavonoids have been studied in certain monocotyledons [14] and eudicotyledons [15,16,17]. These pathways are regulated by a variety of transcription factors (TFs) from the MYB, bHLH, bZIP, WD40, and WRKY families [18,19,20]. Among these TFs, members of the R2R3-MYB family have been shown to regulate the biosynthesis of flavonols and anthocyanins. R2R3-MYB TFs are classified into more than 20 subgroups (SGs) [21,22], which share similar functions [23]. SG5 to SG7 are mostly involved with flavonoid biosynthesis [24,25]. For instance, the *TT2* gene in *Arabidopsis* regulates the biosynthesis of the procyanidin flavonoids after forming a complex with *TT8* (bHLH) and *TTG1* (WD40) [26].

Several R2R3-MYB TFs have been reported to be involved in the regulation of flavonoid biosynthesis in Tartary buckwheat [27,28,29,30]. *FtMYB116* specifically regulates rutin accumulation by directly binding to the flavonoid-3′-hydroxylase (F3′H) promoter region [31], while *FtMYB1* and *FtMYB2* positively regulate the synthesis of procyanidins and anthocyanins [32]. Most of the TF genes mentioned above belong to SG5 R2R3-MYB. Members of SG7 R2R3-MYB have a greater effect on flavonol biosynthesis by regulating the expression of early structural genes [33,34]. Currently, the only member of SG7 R2R3-MYB in Tartary buckwheat, *FtMYB6*, has been demonstrated to regulate the biosynthesis of flavonols in Tartary buckwheat hairy roots and transgenic tobacco by upregulating the expression of *F3H* and *FLS* genes [35]. Our previous study suggested that another SG7 R2R3-MYB gene named *FtMYB163* (*FtPinG0009153900.01*) was also likely to regulate flavonoid biosynthesis in Tartary buckwheat [34]. However, details of FtMYB163 function remain obscure, and it is important to examine the function of this and other members of the SG7 MYB TFs to better understand the biosynthetic pathways of flavonoid synthesis in Tartary buckwheat.

In this study, we analyzed the tissue-specific expression pattern of *FtMYB163* and flavonoid content in Tartary buckwheat. We also over-expressed *FtMYB163* in transgenic *Arabidopsis* and investigated the effect on flavonoid concentrations and the expression of relevant endogenous genes. This study provides valuable information for understanding the function and the potential molecular mechanism of FtMYB163 in flavonoid biosynthesis in Tartary buckwheat.

## 2. Results

### 2.1. Cloning and Molecular Characteristics of FtMYB163

The complete open reading frame (ORF) of *FtMYB163* was isolated from the Tartary buckwheat variety “Heiku013” by PCR using gene-specific primers. The gene was 1119 bp in length and comprised of three exons and two introns (Appendix A). It encoded a 372 amino acid protein with a predicted molecular mass of 42.18 kD and a calculated pI of 6.23 (Appendix A).

Multiple sequence alignment found that FtMYB163 showed 94.09% amino acid identity with FeMYBF1 (LC369592,) from common buckwheat (*F. esculentum*); 44.44% with FtMYB6 (QPC96978); and more than 39% identity with AtMYB12 (AEC10843), CsMYBF1 (KT727073), and VvMYBF1 (FJ948477) (Figure 1). The FtMYB163 protein has R2 and R3 SANT domains at the N terminus, which are involved in sequence-specific DNA binding. In addition, the two SG7 motifs were partially conserved in FtMYB163: motif1 [K/R]R/x][R/K]xGRT[S/x][R/G]xx[M/x]K and motif2 [W/x][L/x]LS [35]. Phylogenetic analysis showed that FtMYB163 clustered with FeMYBF1, AtMYB11, AtMYB12, and FtMYB6 (Figure 2), all of which belong to the SG7 subfamily and participate in the regulation of flavonoid metabolism. These results provide the first indication that FtMYB163 is a member of the SG7 R2R3-MYB subgroup and may have a similar function as FtMYB6, regulating flavonol biosynthesis in Tartary buckwheat.

### 2.2. Subcellular Localization and Transcriptional Activity of FtMYB163

To determine the subcellular localization of the FtMYB163 protein, the GFP reporter gene was fused to the 3′-terminus of the *FtMYB163* coding region and ligated into 16318-hGFP plasmid to generate the FtMYB163-GFP vector. The control vectors 16318-hGFP and FtMYB163-GFP were transiently expressed in tobacco leaves. The fluorescence signal from the unfused GFP control (16318-hGFP) was localized to the cytoplasm and the nucleus, whereas the FtMYB163-GFP fusion protein was localized exclusively to the nucleus (Figure 3A).

We demonstrated that FtMYB163 has transcriptional activity using the yeast hybridization system. As shown in Figure 3B, cells transformed with the pGBKT7 (negative control) could only grow on the SD/-Leu/-Trp medium. By contrast, both the pGBKT7-FtMYB163 and pGBKT7-53 (positive control) could grow normally on SD/-Leu/-Trp and SD/-Ade/-His/-Leu/-Trp selection media. When X-α-gal was included in the media, the cells become blue (Figure 3B). These results indicate that FtMYB163 localizes to the nuclear and has transcriptional activity.

### 2.3. Comparing FtMYB163 Expression and Flavonoid Content in Tartary Buckwheat Tissues

FtMYB163 function was examined by first measuring the expression of *FtMYB163* in the roots (R), stems (ST), leaves (L), and flowers (F) at three stages of plant development (spouting, six-leaf stage, and maturation). Expression was also monitored in seed over a 30 d period during seed maturation. As shown in Figure 4A, *FtMYB163* expression was detected in all these tissues, with the stems and flowers showing slightly higher expression than the other tissues. Expression of *FtMYB163* in developing seed was the highest on day 16 (D16) after grouting, followed by D13 and D20. The lowest level of *FtMYB163* expression in seed occurred at D30 and was only 8% of D16 (Figure 4B).

To investigate the function of FtMYB163 further, we compared the expression of *FtMYB163* with 14 other genes involved in flavonoid biosynthesis in different tissues of Tartary buckwheat. Initially, we used the publicly available transcriptome data for Tartary buckwheat tissues [34,36], specifically the roots (R), stems (ST), leaves (L), flowers (F), and seeds. The seed data covered three development stages: before grouting (S1), at the filling stage (S2), and mature seed (S3). The database showed that the expression of *FtMYB163* and most of the other genes was the highest in the leaves, while the lowest expression was in the S3 seed. The cluster analysis indicated that the expression pattern of *FtMYB163* was similar to 12 of the 14 genes of the flavonoid biosynthesis-related genes, including *DRF* involved in anthocyanin biosynthesis. The two genes not showing the same general pattern were *F3′H-2* and *F3′5′H* (Figure 4C).

We confirmed the *FtMYB163* expression pattern using RT-PCR to measure different Tartary buckwheat tissues, including the stem (ST), the top leaf of adult plants (L1), the top three leaves of adult plants (L2), flowers (F) and three stages of seed development (S1, S2, and S3, as above). The results again showed that *FtMYB163* expression was highly expressed in leaves and relatively low in seeds (Figure 4D).

We then used HPLC to determine how *FtMYB163* expression levels compared with the total flavonoid contents in those same tissues. Flavonoid contents were greatest in the L2 and L1 leaves (52.8 and 39.9 mg/g DW respectively) and lowest in the S1 (20.5 mg/g DW). Contents in the stem and flowers were 36.8 and 24.3 mg/g DW, respectively (Figure 4E). A highly significant positive correlation occurred between *FtMYB163* expression and total flavonoid content in these tissues (*R*^2^ = 0.9042) (Figure 4F). These results indicate that *FtMYB163* expression influences the synthesis of flavonoids in Tartary buckwheat.

### 2.4. Overexpression of FtMYB163 in Arabidopsis Increases Flavonoid Accumulation

To validate the function of *FtMYB163* further, the complete *FtMYB163* ORF was expressed in the *Arabidopsis thaliana* Columbia-0 (Col-0) using the CaMV*35S* promoter (Appendix A). Nine independent homozygous T_3_ lines showing high *FtMYB163* expression were generated and three (OE-2, OE-5, and OE-7) were selected for further analyses (Figure 5A). Total flavonoid contents in the leaves of the transgenic and wild-type (WT) plants are shown in Figure 5B. Transgenic line OE-5 exhibited the highest accumulation of total flavonoids (2.43 mg/g FW), followed by OE-2 (2.21 mg/g FW) and OE-7 (1.96 mg/g FW), and all three were more than five-fold greater than the total flavonoid content in WT plants (0.4 mg/g FW). These results demonstrate that overexpressing *FtMYB163* in *Arabidopsis* significantly increased the total flavonoid content.

We then determined which individual flavonoids showed the greatest changes in the leaves of the transgenic *Arabidopsis* lines. Analysis with LC-MS/MS detected 18 flavonoids that were indicatively higher in the transgenic lines compared with the WT. These included two chalcones, three dihydroflavones, one dihydroflavone glycoside, two dihydroflavonols, two flavones, and eight flavonols (Table 1). The five flavonoids showing the greatest changes were naringenin 7-O-glucoside, eriodictyol, naringenin chalcone, isorhamnetin-3-O-neohespeidoside, and quercetin. The content of these was at least 50-fold greater in one or more of the transgenic lines compared with the WT (Table 1). The flavonols quercetin and kaempferol were 55-fold and 43-fold greater than the WT, respectively. Notably, no changes were detected in the content of proanthocyanidins or anthocyanins. These results suggest that *FTMYB163* expression in *Arabidopsis* alters the biosynthesis of some flavonoids but not anthocyanins.

### 2.5. Expression Analysis of Key Enzyme Genes of Flavonoid Synthesis

We investigated how *FtMYB163* expression in *Arabidopsis* affected the expression of endogenous *Arabidopsis* genes in the flavonoid synthesis pathway. The expression of *AtCHS*, *AtCHI*, *AtF3H*, *AtF3′H*, *AtFLS*, *AtDFR*, and *AtANS* was measured in the leaves of WT and transgenic lines. As shown in Figure 6, *FtMYB163* expression induced the greatest changes in *AtFLS*, *AtCHS,* and *AtCHI* expression, with average increases among the three transgenic lines of 28-fold, 130-fold, and 12-fold (*p* < 0.01), respectively, compared with the WT (Figure 6A–C). Smaller average increases of 2.3-fold were measured for *AtF3H* expression (Figure 6D), while much smaller and variable responses were detected for *AtF3′H*, *AtDFR*, and *AtANS* (Figure 6E–G). These results indicated that *FtMYB163* enhanced the biosynthesis of flavonoids, including flavonols, by altering the expression genes in the flavonoid synthesis pathway.

## 3. Discussion

Flavonoids are important secondary metabolites that benefit plants and human health. Flavonoids are synthesized in plants by the flavonoid biosynthetic pathway, which consists of early flavonoid biosynthesis and late flavonoid synthesis. The early flavonoid biosynthesis pathway mainly depends on the activity of the enzymes CHS, CHI, F3H, F3′H and FLS, which ultimately form flavonols and glycosides [37]. This study investigated the underlying mechanisms of flavonoid biosynthesis in different tissues of developing Tartary buckwheat. We isolated the *FtMYB163* gene, which encodes a predicted protein, with the conserved R2R3 domain and both SG7 motif1 and SG7 motif2 domains. Phylogenetic analysis supported FtMYB163 belonging to the SG7 subgroup of R2R3-MYB transcription factors. These include FeMYBF1 from common buckwheat, FtMYB6 from Tartary buckwheat, and AtMYB12 from *Arabidopsis* (Figure 1 and Figure 2). Previous studies have demonstrated that this group of MYB TFs regulates the synthesis of flavonols [22,38,39]. For example, flavonol biosynthesis in *Arabidopsis* is spatiotemporally regulated by the SG7 MYBs, including MYB11, MYB12, and MYB111 [12]. Yao et al. [35] expressed *FtMYB6* in *Arabidopsis* and found that transcript levels and contents of rutin (a flavonol glycoside) were concentrated in the roots, stems, leaves, and flowers of the transgenic plants. Overexpression of *MtMYB134* in both *Arabidopsis* and hairy roots of *Medicago truncatula* enhanced the biosynthesis of various flavonol derivatives by promoting the early flavonol biosynthesis genes *CHS* and *FLS* [39]. Also, *FtMYB31* expression in transgenic tobacco promoted flavonol biosynthesis by inducing the expression of *CHS*, *F3H*, and *FLS* genes, and the expression of *FtMYB6* in hairy roots and transgenic tobacco significantly increased the accumulation of rutin and other flavonols by inducing the expression of *F3H* and *FLS1* [30,35]. Similar findings have been reported in grape [38], potato [40], common buckwheat [37], *Chrysanthemum morifolium* [41], and *Paeonia qiui* [33]. Therefore, we speculated that *FtMYB163* had a similar function to other genes in the SG7 subgroup. 

We demonstrated that *FtMYB163* expression in different tissues of Tartary buckwheat was positively correlated to the total flavonoid content in those tissues (Figure 4D–F), which is consistent with our preliminary observations [34]. The expression profile of *FtMYB163* in different tissues of Tartary buckwheat was similar to the expression of several structural genes, *CHS*, *CHI*, *F3H*, *F3′H*, and *FLS*, involved with flavonoid synthesis (Figure 4C), which supports the conclusion that *FtMYB163* is involved with the regulation of flavonoid metabolism. 

Some MYB TFs can also suppress the expression of genes in this pathway, thereby affecting flavonoid accumulation in other ways [38,42]. Our study found that expression of *FtMYB163* in *Arabidopsis* enhanced the total flavonoid content of the transgenic lines compared with the WT (Figure 5). In particular, the contents of two flavone and eight flavonol compounds in the transgenic lines were significantly greater than in the WT (Table 1). Furthermore, the relative expression levels of *AtCHS*, *AtCHI*, *AtFLS,* and *AtF3H* genes were consistently greater in all transgenic lines, whereas *AtDFR*, *AtF3′H*, and *AtANS* expression levels were either unchanged, much smaller, or sometimes reduced (e.g., *AtDFR* and *AtANS* expression in OE-7) (Figure 6). Interestingly, the FLS and DFR enzymes usually impact different flavonoid pathways: FLS for the synthesis of colorless flavonols and DFR for the synthesis of colored anthocyanidins [43]. The strong induction of *AtFLS* in all transgenic *Arabidopsis* lines is consistent with the accumulation of eight flavonol compounds in those plants. By contrast, the changes in *AtDFR* expression were much smaller and even suppressed in line OE-7 (Figure 6F,G), which is consistent with there being no detectable changes to the contents of proanthocyanidin and anthocyanin in those lines. These results indicated that *FtMYB163* increased the accumulation of the total flavonoid and promoted flavonol biosynthesis by regulating the expression of *CHS*, *CHI*, *FLS*, and *F3H*. 

Although similar to *FtMYB6*, overexpression of *FtMYB163* could promote the content of flavonoid and flavonol in transgenic plants (Figure 5B and Table 1), the expression of *FtMYB6* and the flavonol accumulation was involved in the light environment [35]. In addition, FtMYB6 and FtMYB116 regulated flavonoid synthesis by binding to the promoters of structural genes *FtF3H*, *FtFLS1*, or *FtF3′H* [31,35]. Therefore, whether FtMYB163 could directly bind to the promoter of the above-mentioned structural genes to regulate flavonol biosynthesis, and whether the expression of *FtMYB163* is affected by environmental factors, such as light, need to be investigated further. Meanwhile, flavonoids can enhance the resistance to abiotic stresses in plants, so we also could further explore the relationship between *FtMYB163* and plant stress resistance.

## 4. Materials and Methods

### 4.1. Plant Materials and Culture Conditions

Seed of the Tartary buckwheat variety “Heiku013” was provided by the Guizhou Normal University (Guiyang, China). Seeds were sown and grown in small pots in a growth chamber under controlled conditions: 16/8 h light/dark cycle at 25 °C, 70% relative humidity, and ~150 μmol m^−2^ s^−1^ light intensity. After growing to the filling stage, different tissues, including roots (R), stems (ST), leaves (L), flowers (F), and seeds (S), were collected, frozen in liquid nitrogen, and then stored at −80 °C for later extraction of total RNA. Tobacco (*Nicotiana benthamiana*) was used for examining subcellular localization of FtMYB163 and stable transformations were generated in *Arabidopsis* using the same methods described previously [44,45].

### 4.2. Gene Cloning and Sequencing

Based on the reference sequence of “Pinku No.1” cDNA, the *FtMYB163* gene was amplified by specific primers (Appendix A). Gene and predicted protein structures were analyzed online using GSDS 2.0 (https://gsds.gao-lab.org/ (accessed on 5 July 2023)) and ExPASy server (https://web.expasy.org/protparam/ (accessed on 5 July 2023)). Amino acid sequences of other related MYB transcription factors (TFs) were collected from NCBI (https://blast.ncbi.nlm.nih.gov/Blast.cgi (accessed on 5 July 2023)), and the phylogenetic analysis was carried out using the Neighbor-Joining (NJ) method of MEGA 11 software. The multiple sequence alignment of chosen amino acids with high identity and similarity was performed with DNAMAN 7.0 software (LynnonBiosoft, San Ramon, CA, USA).

### 4.3. RNA Extraction and Gene Expression Analysis

Total RNA was extracted using the Omega E.Z.N.A Plant RNA Kit (Omega Bio-Tek, Norcross, GA, USA). The quantitative RT-PCR (qRT-PCR) was performed using NovoStart^®^ SYBR q-PCR SuperMix Plus (Novoprotein, Suzhou, China). *FtActin7* and *AtActin2* were used as the Tartary buckwheat and *Arabidopsis* internal reference genes, respectively. The relative expression levels were calculated with the 2^−ΔΔCT^ method [46]. All analyses were performed for a minimum of three biological replicates per sample. The primers used for qRT-PCR are listed in Appendix A.

### 4.4. Subcellular Localization

The *FtMYB163* sequence without terminator sequence was amplified by PCR using primers housing *Sal* I and *BamH* I restriction sites. The ClonExpress^®^ II One Step Cloning kit (Vazyme, Nanjing, China) was used to insert this fragment into the 16318-hGFP plasmid to generate the FtMYB163-GFP vector, where the CaMV*35S* promoter drives expression of the transgene. Subcellular localization was determined by transiently expressing translational fusions of FtMYB163 with GFP in *Nicotiana benthamiana* leaves according to the method described before [47].

### 4.5. Transcriptional Activation of FtMYB163

To verify the transcriptional activation of FtMYB163, the full-length CDS of *FtMYB163* was inserted into the pGBKT7 to construct the pGBKT7-*FtMYB163* recombinant plasmid. The empty pGBKT7 vector (negative control), pGBKT7-*FtMYB163* vector, and pGBKT7-*53* vector (positive control) were introduced into the yeast strain AH109 and grown on SD/-Leu/-Trp and SD/-Ade/-His/-Leu/-Trp selective plates. Positive cells were subsequently grown on SD/-Ade/-His/-Leu/-Trp plates containing X-gal. All plates were incubated in the dark at 28 °C for three to five days and photographed. All primers are listed in Appendix A.

### 4.6. Stable Transformation of Arabidopsis

*FtMYB163* was inserted into the intermediate vector pDONR22 by the Gateway technique (Thermo Fisher Scientific, Waltham, MA, USA). A specific primer with an *attB* site was designed to construct the pK7WG2D-35S-*FtMYB163* vector. The vector was transformed into *Arabidopsis* ecotype Columbia-0 (Col-0) via the floral dip method mediated by *Agrobacterium tumefacien* strain *GV3101* [48]. The first generation (T_1_) seeds of transgenic plants were collected and screened in a 1/2 MS medium with 50 mg/L kanamycin and confirmed by PCR. qRT-PCR was used to select transgenic *Arabidopsis* plants with high *FtMYB163* expression. Independent homozygous T_3_ seeds were collected and used for all following experiments. Primer sequences are listed in Appendix A.

### 4.7. Determination of the Flavonoid Content

The following tissues of “Heiku013” Tartary buckwheat were collected for analyses: Stem, Leaf 1 (top leaf of adult plants), Leaf 2 (top three leaves of adult plants), Seed 1 (seeds before grouting), Seed 2 (seed at filling stage), and Seed 3 (mature seed). Total flavonoid content was determined with the AlCl_3_ method [30]. The standard curve of rutin used was y = 0.002x + 0.0539 (*R*^2^ = 0.9999). Briefly, dry sample powder (0.1 g) was dissolved in 3 mL of 70% methanol solution (MS) and shaken for 4 h at 65 °C protected from light at 160 ± 10 r/min. The sample solution was ultrasonically extracted at 15% output power for 10 min and then centrifuged at 8000 rpm/min for 10 min. The supernatant was collected and filtered through a 0.45 μm microporous membrane, dissolved in 70% MS to 5 mL, and shaken well. After this, the standard rutin solution (1.0 mg/mL) was pipetted at 0.0 mL, 0.5 mL, 1.0 mL, 1.5 mL, and 3.5 mL into different 10-mL volumetric flasks. Then, 2 mL 0.1 mol/L AlCl_3_ and 3 mL 1 mol/L CH_3_COOK were added. Subsequently, 70% MS was used to bring the volume to 10 mL, and it was stewed at room temperature for 30 min. The absorbance at 420 nm was measured, and a standard curve was established based on the absorbance value. Finally, the total flavonoid content was calculated based on the absorbance values of the tested samples and the standard curve.

Flavonoid content in transgenic *Arabidopsis thaliana* plants was determined by gas chromatography-mass spectrometry (GC-MS). Briefly, the frozen samples were freeze-dried, ground for 1.5 min in a grinder under 30 Hz frequency. Then, 20 mg of the ground sample powder was dissolved in 10 μL of a 4000 nM internal standard mixed working solution and 500 μL of 70% methanol extract. The samples were then subjected to an ultrasonic treatment for 30 min and centrifuged at 12,000 r/min at 4 °C. The supernatant was collected and filtered with a 0.22 μm microporous membrane. The final solution (metabolites) was detected by Ultra Performance Liquid Chromatography (UPLC) and Tandem Mass Spectrometry (MS/MS), and the structure of the metabolites was analyzed using the Metware Database. Subsequently, the quantitative analysis of metabolites was measured by the Multiple Reaction Monitoring (MRM) mode of triple quadrupole mass spectrometry. 

The UPLC collection conditions included the chromatographic column: Waters T3 C18 column 1.8 μm, 2.1 mm × 100 mm, mobile phases A (ultrapure water added with 0.05% formic acid) and B (acetonitrile with 0.05% formic acid), 0.35 mL/min flow rate, 40 °C column temperature, 2 μL injection volume, and elution gradient. Mobile phase A/B was 90:10 (*v*/*v*), 80:20, 30:70, 5:95, 5:95, 90:10, and 90:10 at 0 min, 1 min, 9 min, 12.5 min, 13.5 min, 13.6 min, and 15 min, respectively. The MS/MS collection conditions included electrospray ionization (ESI, 550 °C), mass spectrum voltage (5500 V under positive ion mode and −4500 V under negative ion mode), and curtain gas (CUR, 35 psi). Each ion pair was scanned based on declustering potential (DP) and collision energy (CE) optimization in the Q-Trap 6500+ (PerkinElmer, Hopkinton, MA, USA).

### 4.8. Statistical Analysis and Reproducibility

Each experiment included at least five samples and was repeated thrice to confirm reproducibility. Origin software version 2021 (OriginLab Corporation, Northampton, MA, USA) was used to plot the figures. The statistical analysis of the data was conducted using one-way ANOVA, including Tukey’s test, and was performed using the SPSS 21.0 software (IBM, Chicago, IL, USA).

## 5. Conclusions

We conclude that *FtMYB163* encodes an SG7 R2R3-MYB transcription factor. Measurements in Tartary buckwheat and transgenic *Arabidopsis* suggest that expression of this gene regulates the synthesis of flavonoids, especially flavonols, by inducing the expression of the early structural genes *CHS*, *CHI*, *F3H*, and *FLS*. This study lays the foundation for future attempts to improve the nutritional value of Tartary buckwheat.

## Figures and Tables

**Figure 1 plants-13-02704-f001:**
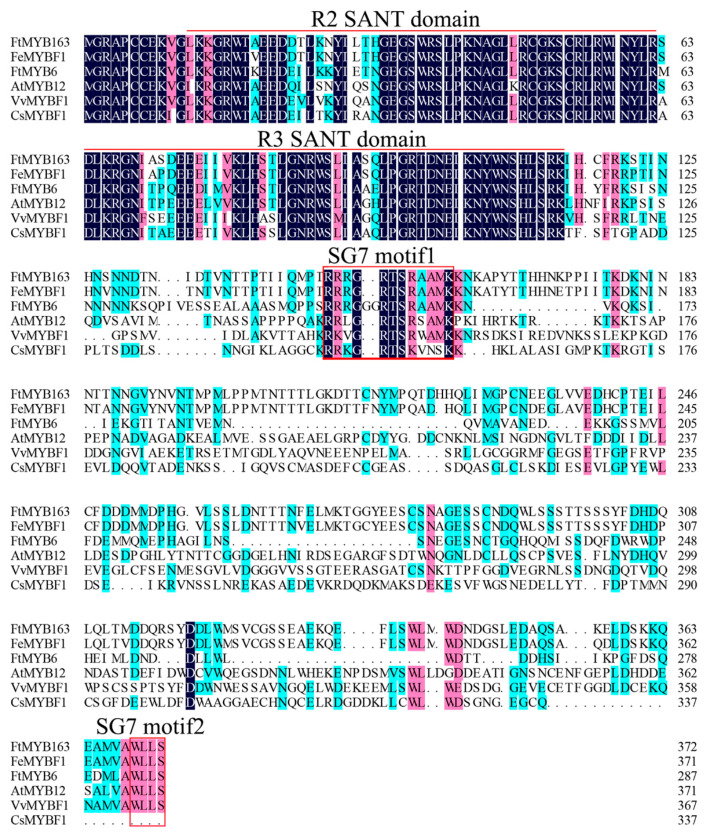
Multiple sequence alignment of FtMYB163. The transcription factors were FeMYBF1 (LC369592) from common buckwheat, FtMYB6 (QPC96978) from Tartary buckwheat, AtMYB12 (AEC10843) from Arabidopsis, VvMYBF1 (FJ948477) from grape, and CsMYBF1 (KT727073) from citrus. Identical (100%), conservative (75–99%), and blocks (50–74%) of similar amino acid residues are shaded in deep blue, cherry red, and cyan, respectively. The R2/R3 SANT domain and SG7 motif1/2 are indicated in the red line and red box, respectively.

**Figure 2 plants-13-02704-f002:**
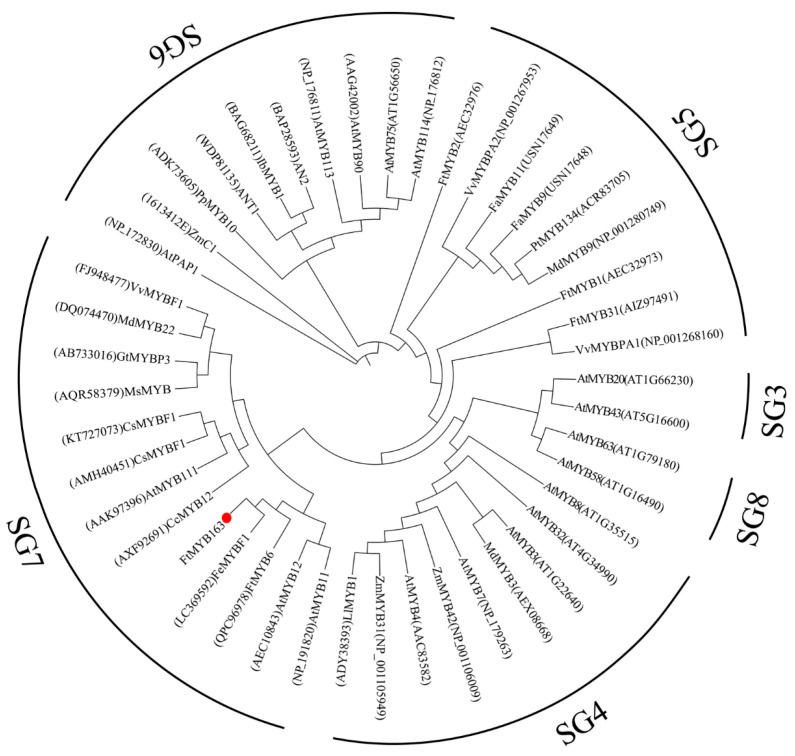
Phylogenetic analysis of FtMYB163. GeneBack accession numbers are listed as follows: AtMYB58 (NP_173098), AtMYB63 (NP_001321204), LlMYB1 (ADY38393), ZmMYB31 (NP_001105949), ZmMYB42 (NP_001106009), MdMYB3 (AEX08668), AtMYB7 (NP_179263), AtMYB4 (AAC83582), AtMYB8 (NP_849749), AtMYB32 (NP_195225), AtMYB3 (NP_564176), FtMYB2 (AEC32976), FaMYB11 (USN17649), FaMYB9 (USN17648), MdMYB9 (NP_001280749), PtMYB134 (ACR83705), VvMYBPA1 (NP_001268160), VvMYBPA2 (NP_001267953), PpMYB10 (ADK73605), AN2 (BAP28593), ANT1 (WDP81135), IbMYB1 (BAG68211), AtMYB113 (NP_176811), AtMYB114 (NP_176812), GtMYBP3 (AB733016), MdMYB22 (DQ074470), AtMYB111 (AAK97396), AtMYB11 (NP_191820), MsMYB (AQR58379), CcMYB12 (AXF92691), AtMYB90 (AAG42002), AtPAP1 (NP_172830), AtMYB75 (NP_176057), AtMYB43 (NP_197163), AtMYB20 (NP_176797), FtMYB1 (AEC32973), FtMYB31 (AIZ97491). FtMYB163 is highlighted with a red dot.

**Figure 3 plants-13-02704-f003:**
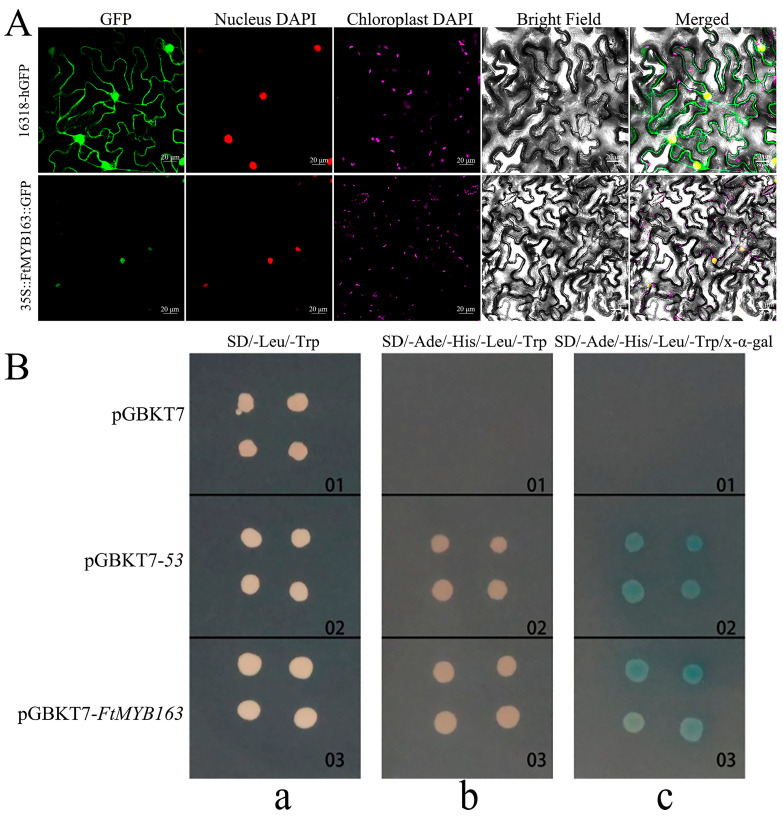
Subcellular localization and transcription activation activities of FtMYB163. (**A**) Subcellular localization of FtMYB163-GFP fusion protein in *Nicotiana benthamiana* leaves. GFP: Green fluorescent protein; DAPI: 4′,6-diamidino-2-phenylindole stain; 16318-hGFP was used as the control. Scale bar: 20 μm. (**B**) Transcription activation analysis of FtMYB163 in yeast AH109 cells. The transformed cells were plated on an (**a**) SD/-Leu/-Trp, (**b**) SD/-Ade/-His/-Leu/-Trp, and (**c**) SD/-Ade/-His/-Leu/-Trp/x-α-gal medium and incubated in an incubator at 30 °C for 3~5 d. pGBKT7, negative control; pGBKT7-*53*, positive control.

**Figure 4 plants-13-02704-f004:**
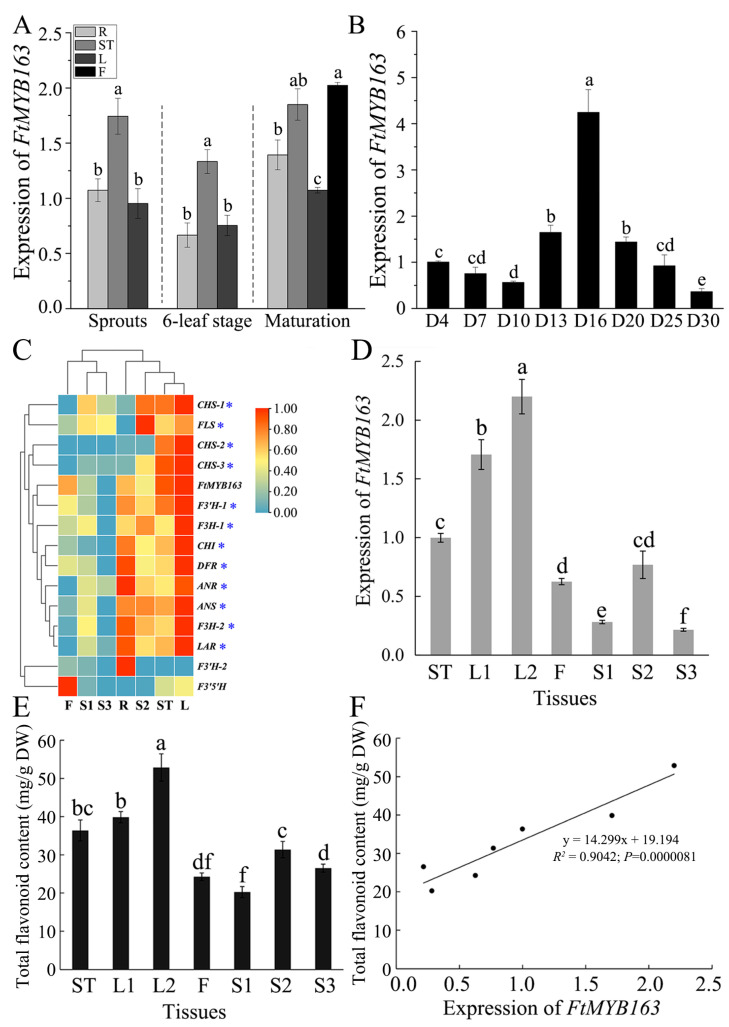
Expression pattern of *FtMYB163* and the total flavonoid content in Tartary buckwheat. The expression of *FtMYB163* in (**A**) R (roots), ST (stems), L (leaves), and F (flowers) at sprouts, six-leaf stage, and maturation stage and (**B**) in seeds at different developmental stages. (**C**) Gene expression clustering heat map of *FtMYB163* and Tartary buckwheat flavonoid biosynthesis structure. (**D**) *FtMYB163* expression analysis and (**E**) the total flavonoid content detection in each tissue at the corresponding period (as in (**C**)). (**F**) The correlation between the *FtMYB163* expression in different tissues and the content of total flavonoids. D4~D30, the seeds 4~30 days after flowering; L1, top one leaf of adult plants; L2, top three leaves of adult plants; S1, seeds before grouting; S2, seeds at filling stage; S3, mature seeds. Blue asterisks indicated the structural genes co-expressed with *FtMYB163* during the three seed development periods. The expression levels were evaluated by the 2^−ΔΔCT^ method, and *FtActin7* was used as a reference gene. The values are represented as mean ± SD (n = 5) and marked with different letters to indicate statistically significant differences at *p* < 0.05 (Tukey’s test).

**Figure 5 plants-13-02704-f005:**
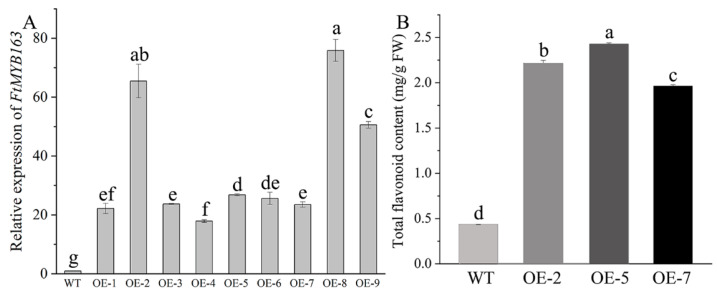
Characterization of T_3_ transgenic *Arabidopsis* lines expressing *FtMYB163.* (**A**) Relative expression levels of *FtMYB163* in independent homozygous T_3_ lines using qRT-PCR. *AtACT2* was used as a reference gene. Data show the mean ± SD of three biological replicates. Different letters indicate significant differences at *p* < 0.05 (Student’s *t*-test). (**B**) Total flavonoid content of WT *Arabidopsis* and three transgenic T_3_ lines expressing *FtMYB163* (OE-2, OE-5, and OE-7). Values are mean ± SD (n = 10), and different letters indicate significant differences at *p* < 0.01 (Tukey’s test).

**Figure 6 plants-13-02704-f006:**
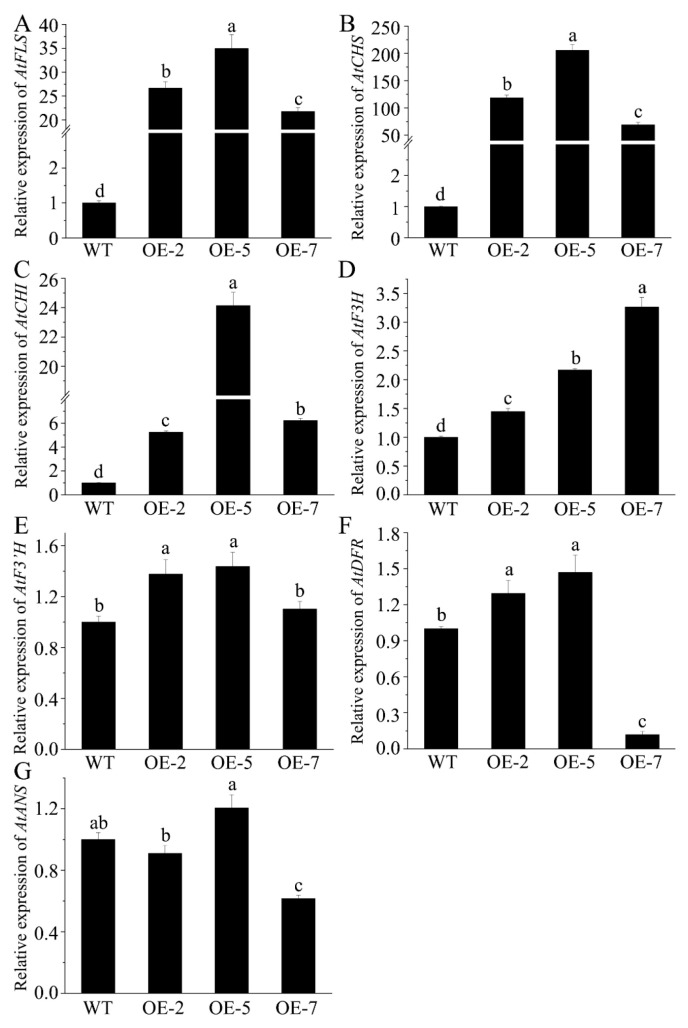
Assessing how the expression of *FtMYB163* in *Arabidopsis* leaves affects the expression of endogenous genes in the flavonoid synthesis pathway. The relative expression of Arabidopsis genes (**A**) flavonol synthase (*AtFLS*), (**B**) chalcone synthase (*AtCHS*), (**C**) chalcone isomerase (*AtCHI*), (**D**) flavonoid 3-hydroxylase (*AtF3H*), (**E**) flavonoid 3′-hydroxylase (*AtF3′H*), (**F**) dihydroflavonol 4-reductase (*AtDFR*), and (**G**) anthocyanidin reductase (*AtANS*) were measured in WT and transgenic lines (OE-2, OE-5, and OE-7). Gene expression in the WT was set to 1.0 to provide fold changes in expression. The *AtACT2* gene was used as a reference. Values represent mean ± SD (n = 5), and different letters indicate significant differences (*p* < 0.01) (Tukey’s test).

**Table 1 plants-13-02704-t001:** The fold change in flavonoid content between WT and transgenic *Arabidopsis*.

No.	Class	Compounds	WT	OE-2	OE-5	OE-7
1	Chalcone	Trilobatin	1 ± 0.1 c	5.4 ± 0.1 b	7.8 ± 1.1 a	2.7 ± 0.2 c
2	Chalcone	Naringenin chalcone	1 ± 0.2 b	84.8 ± 9.9 a	84.3 ± 15.5 a	25.1 ± 2.0 b
3	Dihydroflavone	Hesperidin	1 ± 0.1 b	11.7 ± 2.7 a	15.4 ± 0.2 a	14.8 ± 0.1 a
4	Dihydroflavone	Eriocitrin	1 ± 0.2 b	39.8 ± 7.9 a	37.7 ± 2.0 a	16.3 ± 2.6 b
5	Dihydroflavone glycoside	Naringenin-7-O-glucoside	1 ± 0.0 b	246.3 ± 20.1 a	256.4 ± 50.3 a	61.8 ± 7.6 b
6	Dihydroflavone	Eriodictyol	1 ± 0.1 b	134.4 ± 27.7 a	157.5 ± 43.5 a	14.6 ± 4.3 b
7	Dihydroflavonol	Dihydrokaempferol	1 ± 0.2 b	3.9 ± 1.2 ab	6.2 ± 2.1 a	1.3 ± 0.1 b
8	Dihydroflavonol	Taxifolin 7-O-rhamnoside	1 ± 0.1 c	30.1 ± 4.2 a	26.2 ± 1.1 a	14.2 ± 1.3 b
9	Flavone	Luteolin-7-glucoside	1 ± 0.0 b	13.1 ± 13.1 a	4.5 ± 0.3 ab	4.2 ± 2.1 ab
10	Flavone	Apigenin 7-glucoside	1 ± 0.1 b	37.7 ± 4.7 a	39.5 ± 4.5 a	14.6 ± 2.3 b
11	Flavonol	Afzelin/kaempferol-3-rhamnoside	1 ± 0.0 c	3.0 ± 0.2 b	9.0 ± 0.1 a	1.5 ± 0.1 c
12	Flavonol	Isorhamnetin-3-O-glucoside	1 ± 0.2 b	19.0 ± 0.0 b	71.2 ± 14.5 a	4.6 ± 1.6 b
13	Flavonol	Astragalin/kaempferol-3-glucoside	1 ± 0.0 b	1.6 ± 0.0 b	5.5 ± 0.3 a	1.2 ± 0.2 b
14	Flavonol	Quercitrin/Quercetin 3-rhamnoside	1 ± 0.0 c	2.5 ± 0.4 b	8.5 ± 0.1 a	1.2 ± 0.1 c
15	Flavonol	Baimaside	1 ± 0.3 b	3.5 ± 0.3 ab	6.2 ± 2.0 a	2.8 ± 0.3 ab
16	Flavonol	Avicularin/Quercetin 3-alpha-L-arabinofuranoside	1 ± 0.0 b	2.1 ± 0.1 b	12.6 ± 0.8 a	1.3 ± 0.2 b
17	Flavonol	Quercetin	1 ± 0.2 b	7.0 ± 3.1 b	55.5 ± 5.1 a	1.9 ± 0.0 b
18	Flavonol	Kaempferol	1 ± 0.4 b	28.3 ± 4.1 a	42.6 ± 7.5 a	2.2 ± 0.3 b

Note: Data represent mean ± SD of two biological replicates. Lowercase letters indicate significant differences at *p* < 0.05 (Tukey’s test).

## Data Availability

Data are available on request to the corresponding author’s email with appropriate justification.

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
