# Peer review of "FtMYB163 Gene Encodes SG7 R2R3-MYB Transcription Factor from Tartary Buckwheat (Fagopyrum tataricum Gaertn.) to Promote Flavonol Accumulation in Transgenic Arabidopsis thaliana"

_plants, 2024, doi:10.3390/plants13192704_

Round 1
Reviewer 1 Report
Comments and Suggestions for Authors
The manuscript by Du et al. is dedicated to the analysis of the structure and the expression of a transcription factor from the MYB family, which is involved in the regulation of flavonoid biosynthesis in Fagopyrum tataricum. For quite a long time this plant species has attracted the attention of plant biologists due to its rich metabolite content and nutritional value. In this study, solid experimental methods have been applied to estimate the expression profile of FtMYB163 gene in various organs of F. tataricum and to create transgenic Arabidopsis expressing FtMYB163. The authors also analyzed the changes in the flavonoid content and identified the flavonoid species in F. tataricum and transgenic Arabidopsis.
There are major and minor comments.
Major comments:
1. Many papers have already been published about this transcription family; therefore, the novelty of this research is not clear. Buckwheat, and specifically Tartary buckwheat, is not a new plant species, and specifically transcription factor R2R3-MYB has been well studied, as indicated by the references and a quick search in Pubmed. The authors should provide a clear novelty of their present study.
2. The authors did not perform a standard comparison of the usual parameters of root growth, leaves, and seed viability etc. in WT and transgenic plants. Most often, when this is the first paper about a new transgenic line that has not yet been published, they try to give the usual growth factors to show that the line is viable. I am confused with Supplementary Fig 1, it is not clear where WT and transgenic plants. Are wild plants phenotypically different from transgenic ones?
3. Re: Fig 4
Is it really possible to make a “correlation” between one gene expression and total flavonoid content? It is quite doubtful that such a multi-stepped pathway as biosynthesis of flavonoids depends only on the expression of one gene encoding one specific transcription factor.
4. Re: The values are represented as mean ± SD (n = 5) and marked with different letters to indicate statistically significant differences at P < 0.05 (Tukey’s test).
Since these are transgenic plants, the statistics must be calculated not using two way ANOVA but ideally using Dunnett’s criterion. Citation from the internet: If you have several groups you want to compare against each other, you should use a post-hoc test. Tukey's HSD is quite common for all-pairwise comparisons, Dunnett's procedure is ideal for multiple-to-one comparisons (several treatments vs. the same control, but not treatment A vs. treatment B).
5. Re: Data represent mean ± SD of two biological replicates. Lowercase letters indicate significant differences at P < 0.05 (Student’s test).
Two biological replicates are not enough and statistics for such a number of replicates may be incorrect.
6. L314-315 …Based on the reference sequence of ‘Pinku No.1’ cDNA, the FtMYB163 gene was amplified by specific primers (Table S2).
What is Pinku No.1? I could not find this in the NCBI database. The sequence number and where it was taken from should be given, and the IDs of all genes should also be indicated.
7. L327-328 ….q-PCR SuperMix Plus (Novoprotein, Suzhou, China). FtActin7 and AtActin2 were used as the Tartary buckwheat and Arabidopsis reference genes, respectively.
For qPCR, only one reference gene was used. Taking into account the fact that the manuscript places a great emphasis on data obtained using quantitative PCR, the authors should have used 2-3 reference genes to make the data more convincing.
8. L377 4.8. Statistical analysis and reproducibility
Nothing is said about what ANOVA the authors used. If there was a comparison between the wild type and the transgenic plants in the time course of during plant growth “leaf, etc.”? There should be a double comparison, since there is a multifactorial comparison. I am not sure if it is correct to use the Student's test for such stats.
9. Obviously, the Methods section should give methodological description detailed enough for the readers to be able to reproduce experiments. Therefore, brief description of the determination of flavonoid content using AlCl3 assay should be given. Moreover, some details of GC-MS protocol and the settings applied should be given.
Minor comments
1. English should be improved, few typos found.
2. L350 FtMYB163 was inserted into the intermediate vector pDONR22 by the Gateway tech… Please indicate the vector manufacturer.
3. Please make sure that whether the content (Results) or concentration (Introduction) of flavonoids is presented.
Comments on the Quality of English LanguageEnglish should be improved.
Author Response
Responses to reviewers’ comments
#Reviewer 1
Major comments
- Many papers have already been published about this transcription family; therefore, the novelty of this research is not clear. Buckwheat, and specifically Tartary buckwheat, is not a new plant species, and specifically transcription factor R2R3-MYB has been well studied, as indicated by the references and a quick search in Pubmed. The authors should provide a clear novelty of their present study.
Reply: Thank you. As you mentioned, there are many studies on this transcription family. In Tartary buckwheat, studies on MYB TFs are mainly divided into two categories: 1) studies on MYB and plant stress resistance, such as drought, cold and salt stress (Gao et al., 2016; Zhou et al., 2015), 2) studies on the regulation of flavonoid biosynthesis (Bai et al., 2014; Luo et al., 2018; Yao et al., 2020). However, the research on the regulation of flavonoid synthesis by MYB mainly focus on the synthesis of anthocyanin and proanthocyanidin (Bai et al., 2014; Luo et al., 2018; Wang et al., 2022), flavonols (including rutin and quercetin) are very important secondary metabolites in Tartary buckwheat. We pay more attention to the role of MYB TFs in regulation of flavonol biosynthesis. Currently, FtMYB116 (Zhang et al., 2019), FtMYB6 (Yao et al., 2020), FtMYB16 (Li et al., 2019) and FtMYB31 (Sun et al., 2019) have been reported to be associated with flavonol synthesis, whereas only FTMYB6 belongs to the SG7 subfamily and regulates flavonol synthesis (Yao et al., 2020). FtMYB163, a novel SG7-MYB TF is confirmed a key regulator of flavone and flavonol biosynthesis, which is closely positively related to flavanones and flavonols (Li et al., 2019), our results further indicated that overexpression of FtMYB163 improved the flavonoid (including naringenin-7-O-glucoside, eriodictyol, and naringenin chalcone) and flavonol (Quercetin and Kaempferol) content in Arabidopsis. This is different from FtMYB6. In addition, the genes involved in flavonoid synthesis regulated by FtMYB6 and FtMYB163 are also different.
Reference:
Gao, F., Yao, H.P., Zhao, H.X., Zhou, J., Luo, X.P., Huang, Y.J., Li, C.L., Chen, H., Wu, Q. Tartary buckwheat FtMYB10 encodes an R2RE-MYB transcription factor that acts as a novel negative regulator of salt and drought response in transgenic Arabidopsis. Plant Physiol Biochem 2016, 109: 387-396.
Zhou, M.L., Wang, C.L., Qi, L.P., Yang, X.B., Sun, Z.M., Tang, Y., Tang, Y.X., Shao, J.R., Wu, Y.M. Ectopic expression of Fagopyrum tataricum FtMYB12 improves cold tolerance in Arabidopsis thaliana. J Plant Growth Regul 2015, 34: 362-371.
Bai. Y.C., Li, C.L., Zhang, J.W., Li, S.J., Luo. X.P., Yao, H.P., Chen, H., Zhao, H.X., Park, S.U., Wu, Q. Characterization of two tartary buckwheat R2R3-MYB transcription factors and their regulation of proanthocyanidin biosynthesis. Physiol Plantarum 2014, 152: 431-440.
Luo, X.P., Zhao, H.X., Yao, P.F., Li, Q.Q., Huang, Y.J., Li, C.L., Chen, H., Wu, Q. An R2R3-MYB transcription factor FtMYB15 involved in the synthesis of anthocyanin and proanthocyanidins from Tartary buckwheat. J Plant Growth Regul 2018, 37: 76-84.
Yao, P., Huang, Y., Dong, Q., Wan, M., Wang, A., Chen, Y., Li, C., Wu, Q., Chen, H., Zhao, H. FtMYB6, a light-induced SG7 R2R3-MYB transcription factor, promotes flavonol biosynthesis in Tartary buckwheat (Fagopyrum tataricum). J Agric Food Chem 2020, 68, 13685-13696.
Wang, L., Deng, R.Y., Bai, Y.C., Wu, H.L., Li, C.L., Wu, Q., Zhao, H.X. Tartary buckwheat R2R3-MYB gene FtMYB3 negatively regulates anthocyanin and proanthocyanin biosynthesis. Int J Mol Sci 2022, 23: 2775.
Zhang, D., Jiang, C. L., Huang, C. H., Wen, D., Lu, J. N., Chen, S., Zhang, T. Y., Shi, Y. H., Xue, J. P., Ma, W., Xiang, L., Sun, W., Chen, S. L. The light-induced transcription factor FtMYB116 promotes accumulation of rutin in Fagopyrum tataricum. Plant Cell Environ 2019, 42, 1240-1351.
Li, J.B., Zhang, K.X., Meng, Y., Li, Q., Ding, M.Q., Zhou, M.L. FtMYB16 interacts with Ftimportin-α1 to regulate rutin biosynthesis in tartary buckwheat. Plant Biotechnol J 2019, 27: 1479-1481.
Sun, Z., Linghu, B., Hou, S., Liu, R., Wang, L., Hao, Y., Han, Y., Zhou, M., Liu, L., Li, H. Tartary buckwheat FtMYB31 gene encoding an R2R3-MYB transcription factor enhances flavonoid accumulation in Tobacco. J Plant Growth Regul 2019, 39, 564-574.
Li, H., Lv, Q., Ma, C., Qu, J., Cai, F., Deng, J., Huang, J., Ran, P., Shi, T., Chen, Q. Metabolite profiling and transcriptome analyses provide insights into the flavonoid biosynthesis in the developing seed of Tartary buckwheat (Fagopyrum tataricum). J Agric Food Chem 2019, 67, 11262-11276.
- The authors did not perform a standard comparison of the usual parameters of root growth, leaves, and seed viability etc. in WT and transgenic plants. Most often, when this is the first paper about a new transgenic line that has not yet been published, they try to give the usual growth factors to show that the line is viable. I am confused with Supplementary Fig 1, it is not clear where WT and transgenic plants. Are wild plants phenotypically different from transgenic ones?
Reply: Thank you. As shown in Fig. S2A, B and Fig. 5A, the transgenic lines were screened in 1/2 MS medium with kanamycin and confirmed by PCR and qRT-PCR. Since FtMYB163 is a gene involved in regulating the synthesis of flavonoids, however, flavonoids have no color compared with anthocyandins (Davies et al., 2003), there is no significant difference between WT and transgenic lines in leaf and root growth (Fig. S2D). Therefore, we compared WT and transgenic lines by measuring flavonoid content (Table 1).
Reference:
Davies, K. M., Schwinn, K. E., Deroles, S. C., Manson, D. G., Lewis, D. H., Bloor, S. J., Bradley, J. M. Enhancing anthocyanin production by altering competition for substrate between flavonol synthase and dihydroflavonol 4-reductase. Euphytica 2003, 131, 259-268.
- Fig 4. Is it really possible to make a “correlation” between one gene expression and total flavonoid content? It is quite doubtful that such a multi-stepped pathway as biosynthesis of flavonoids depends only on the expression of one gene encoding one specific transcription factor.
Reply: Thank you for your advice. In this study, the expression pattern of FtMYB163 was similar to the flavonoid biosynthesis-related gene. Meanwhile, the main flavone of Tartary buckwheat is flavonol including rutin, quercetin, and pyrolycoside, which accounting for almost more than 90% of the total flavonoids of Tartary buckwheat (Li et al., 2022; Ren, 2017). We hypothesized that FtMYB163 is a key gene regulating flavonol synthesis or its expression affects the activity of flavonol synthase. This conjecture has a similar application in FtMYB6 (Yao et al., 2020; Fig. S3).
Reference:
Li, C.L., Yang, J.J., Yang, K., Wu, H.L., Chen, H., Wu, Q., Zhao, H.X. Tartary buckwheat FtF3’H1 as a metabolic branch switch to increase anthocyanin content in transgenic plant. Front Plant Sci 2022, 13: 959698.
Ren, Q. To research progress in chemical constituents and biological activities of Fagopyrun tataricum (L.) Gaertn. J Jining Med Univ (in Chinese) 2017, 40: 251-255.
Yao, P., Huang, Y., Dong, Q., Wan, M., Wang, A., Chen, Y., Li, C., Wu, Q., Chen, H., Zhao, H. FtMYB6, a light-induced SG7 R2R3-MYB transcription factor, promotes flavonol biosynthesis in Tartary buckwheat (Fagopyrum tataricum). J Agric Food Chem 2020, 68, 13685-13696.
- The values are represented as mean ± SD (n = 5) and marked with different letters to indicate statistically significant differences at P < 0.05 (Tukey’s test).
Since these are transgenic plants, the statistics must be calculated not using two way ANOVA but ideally using Dunnett’s criterion. Citation from the internet: If you have several groups you want to compare against each other, you should use a post-hoc test. Tukey's HSD is quite common for all-pairwise comparisons, Dunnett's procedure is ideal for multiple-to-one comparisons (several treatments vs. the same control, but not treatment A vs. treatment B).
Reply: Thank you for your professional advice. We tried to apply Dunnett’s test to Fig.4 A and B using the method you suggested and found that the significant difference obtained by this method was basically the same as Tukey’s test.
However, the significance of Dunnett’s test can only be compared with the same control (In Fig. A and B, we set R and D4 as the control, respectively.), can’t compare the differences between each data. In addition, many studies have used Tukey’s test to perform similar statistical analysis (Yan et al., 2021; Zhang et al., 2023), whether for gene expression or physiological measures. Therefore, to better account for the significance of differences between these data, we still used Tukey’ test.
Reference:
Yan, X.X., Huang, Y., Song, H., Chen, F., Geng, Q.L., Hu, M., Zhang, C., Wu, X., Fan, T.T., Cao, S.Q. A MYB-MAN3-mannose-MNB1 signaling cascade regulates cadmium tolerance in Arabidopsis. Plos Genetics 2021, 17(6): e1009636.
Zhang, J.S., Zhang, X.L., Jia, M.G., Fu, Q., Guo, Y.S., Wang, Z.H., Kong, D.J., Lin, Y.C., Zhao, D.G. Two novel transporters NtNRAMP6a and NtNRAMP6b are involved in cadmium transport in tobacco (Nicotiana tabacum L.). Plant Physiol Bioch 2023, 202: 107953.
- Data represent mean ± SD of two biological replicates. Lowercase letters indicate significant differences at P < 0.05 (Student’s test).
Two biological replicates are not enough and statistics for such a number of replicates may be incorrect.
Reply: We have changed ‘significantly’ to ‘indicatively’ in manuscript R1_with trace track. The sample we used to test the flavonoid content in Table 1 for each biological replicate was relatively large (≥15), and the results trend was consistent. Therefore, although we had only two biological replicates, the results suggest that these differences are potentially indicative. Besides, we also revised ‘Student’s test’ to ‘Tukey’s test’, because we used multiple comparisons instead of simple comparisons between two data.
- L314-315 …Based on the reference sequence of ‘Pinku No.1’ cDNA, the FtMYB163 gene was amplified by specific primers (Table S2).
What is Pinku No.1? I could not find this in the NCBI database. The sequence number and where it was taken from should be given, and the IDs of all genes should also be indicated.
Reply: ‘Pinku No.1’ is the first whole genome sequencing of Tartary buckwheat produced by Shanxi Academy of Agricultural Science (Zhang et al., 2017). The genome sequencing results were published on MBKBASE (https://mbkbase.org/Pinku1/). We also added the gene IDs in Table S2.
Reference:
Zhang, L., Li, X., Ma, B., Gao, Q., Du, H., Han, Y., Li, Y., Cao, Y., Qi, M., Zhu, Y., Lu, H., Ma, M., Liu, L., Zhou, J., Nan, C., Qin, Y., Wang, J., Cui, L., Liu, H., Liang, C., Qiao, Z. The Tartary buckwheat genome provides insights into rutin biosynthesis and abiotic stress tolerance. Mol Plant 2017, 10: 1224-1237.
- L327-328 ….q-PCR SuperMix Plus (Novoprotein, Suzhou, China). FtActin7 and AtActin2 were used as the Tartary buckwheat and Arabidopsis reference genes, respectively.
For qPCR, only one reference gene was used. Taking into account the fact that the manuscript places a great emphasis on data obtained using quantitative PCR, the authors should have used 2-3 reference genes to make the data more convincing.
Reply: Thanks a lot for your suggestion. We agree that more reference genes are preferred. In fact, before we selected them (FtH3 and FtActin7) as internal reference genes, we had already screened them. For example, in Tartary buckwheat, we found the expression levels were almost stable when we used both FtH3 and FtActin7 as reference gene to analyze the expression levels of different tissues and under different conditions, respectively. Therefore, we only used one internal reference gene in subsequent experiments. And the use of this one gene is still useful and reflected the real trends.
- L377 4.8. Statistical analysis and reproducibility
Nothing is said about what ANOVA the authors used. If there was a comparison between the wild type and the transgenic plants in the time course of during plant growth “leaf, etc.”? There should be a double comparison, since there is a multifactorial comparison. I am not sure if it is correct to use the Student's test for such stats.
Reply: Thank you for your professional suggestion. In this study, statistical analysis of data was conducted using one-way ANOVA.
Since there is no significant difference between transgenic plants and WT in morphology, we didn’t compare this during plant growth, only the species and content of flavonoids were detected.
We agree with you that multiple comparisons are used when comparing significant differences, while Student’s test is only used when comparing differences between two data in this study, to better describe the statistical analysis, we have revised this part in the manuscript R1_with trace track (see L441-443).
- Obviously, the Methods section should give methodological description detailed enough for the readers to be able to reproduce experiments. Therefore, brief description of the determination of flavonoid content using AlCl3 assay should be given. Moreover, some details of GC-MS protocol and the settings applied should be given.
Reply: We added the description as your suggestion, please check the detail in manuscript R1_with trace track (see L402-414 and L427-43).
Minor comments
- English should be improved, few typos found.
Reply: We have read the whole manuscript carefully and revised some of our writings. For details, please see the “manuscript R1_with trace track”.
- L350 FtMYB163 was inserted into the intermediate vector pDONR22 by the Gateway tech… Please indicate the vector manufacturer.
Reply: We added this information to the manuscript R1_with trace track (see L388).
- Please make sure that whether the content (Results) or concentration (Introduction) of flavonoids is presented.
Reply: Thank you for your advice. ‘Content’ refers to the amount of something in the leaf or roots or whole plant etc. By contrast, ‘concentration’ refers to the amount of something per unit of volume of extract (e.g. ml or L etc) or per unit weight of dry tissue (e.g. has units like: mg/g shoot dry weight etc). It is also suggested that ‘a concentration is an amount of any type per volume of liquid or gas system, whereas content is an amount of any type per mass of liquid or gas or solid system’ (Fuentes-Arderiu, 2013). However, the concepts are still ambiguous because, depending on the type of amount of the component per volume or mass of a system, there are different types of concentrations and contents. In ‘Introduction’, ‘We also over-expressed FtMYB163 in transgenic Arabidopsis and investigated the effect on flavonoid concentrations and the expression of relevant endogenous genes’ is to emphasize that the flavonoid measurement samples were the tissue extract, as the unit used in the results was mg/g DW or mg/g FW. In Results, we used ‘content’ to describe the detection result, which indicated the amount of flavonoid in the samples (dry weight or fresh weight) we used.
Reference:
Fuentes-Arderiu, X. Concentration and content. Biochemia medica 2013, 23(2): 141-142.

Reviewer 2 Report
Comments and Suggestions for Authors
Dear Authors,
The introduction is specific and very well written, but I think that a concise statement of objectives would strengthen the introduction and better connect the introduction with the rest of the manuscript. The text refers to various studies, which are appropriately selected and cited, but it would be very valuable to compare the results presented in this manuscript with the results of previous studies cited in the text. For example, the authors could better place the significance of FtMYB163 in a broader context, drawing attention to new observations that have emerged in this work. Raising in the discussion topics how the findings from this manuscript could be applied to agriculture or medicine would add value. A second aspect for discussion could be the potential applications or broader effects of manipulating FtMYB163 in crops or other plant species.
Is it known, and could it be presented in the paper, how FtMYB163 interacts with other transcription factors or signaling pathways in Tatar buckwheat? This topic could indicate further research prospects, further studies of the regulatory network involving FtMYB163 or exploration of its role in different environmental conditions.
Author Response
Responses to reviewers’ comments
#Reviewer 2
The introduction is specific and very well written, but I think that a concise statement of objectives would strengthen the introduction and better connect the introduction with the rest of the manuscript. The text refers to various studies, which are appropriately selected and cited, but it would be very valuable to compare the results presented in this manuscript with the results of previous studies cited in the text. For example, the authors could better place the significance of FtMYB163 in a broader context, drawing attention to new observations that have emerged in this work. Raising in the discussion topics how the findings from this manuscript could be applied to agriculture or medicine would add value. A second aspect for discussion could be the potential applications or broader effects of manipulating FtMYB163 in crops or other plant species.
Is it known, and could it be presented in the paper, how FtMYB163 interacts with other transcription factors or signaling pathways in Tatar buckwheat? This topic could indicate further research prospects, further studies of the regulatory network involving FtMYB163 or exploration of its role in different environmental conditions.
Reply: Thank you for your professional suggestion. We have made changes in ‘Introduction’ and ‘Discussion’. Please check it in the manuscript R1_with trace track (L54, L63-64, L83, and L328-L337). We will further focus on the role of FTMYB163 in different environmental conditions and it interacts with other transcription factors or signaling pathways in Tartary buckwheat.

Round 2
Reviewer 1 Report
Comments and Suggestions for Authors
The changes made by the authors in response to my comments are satisfactory and helped to improve the manuscript.